# Adding Low-Dose Propofol to Limit Anxiety during Target-Controlled Infusion of Remifentanil for Gastrointestinal Endoscopy: Respiratory Issues and Safety Recommendations

**DOI:** 10.3390/medicina58091285

**Published:** 2022-09-15

**Authors:** Cyrus Motamed, Frederique Servin, Valerie Billard

**Affiliations:** 1Department of Anesthesia, Institut Gustave Roussy, 94805 Villejuif, France; 2Hôpital Bichat, APHP, 75018 Paris, France

**Keywords:** procedural sedation, propofol, remifentanil, gastrointestinal endoscopy, drug interactions, spontaneous ventilation, target-controlled infusion

## Abstract

*Background**and Objectives*: Remifentanil-based sedation is one of many protocols proposed for endoscopy procedures in spontaneous ventilation, alone or in combination with propofol. However, the effect of these small doses of propofol on the efficacy and safety of remifentanil target-controlled infusion (TCI) deserves to be examined in this context. The objective of this study was to assess the adverse respiratory and cardiovascular effects of small boluses of propofol combined with remifentanil, in comparison with remifentanil alone, and balanced with the quality of sedation and recovery. *Materials and**Methods*: This was an observational bicenter study, representing a subgroup of a larger study describing remifentanil-based procedural sedation. In center 1, patients scheduled for gastrointestinal (GI) endoscopy had remifentanil TCI alone. In center 2, patients had a 10 mg propofol bolus before TCI and other boluses were allowed during the procedure. Remifentanil TCI was started at a target of 2 ng/mL then adapted by 0.5 ng/mL steps according to patient response to endoscopy stimulations. *Results*: Center 1 included 29 patients, while center 2 included 60 patients. No difference was found in the patients’ characteristics, incidence of success, average remifentanil consumption, or cardiovascular variables. Light sedation was achieved when propofol was added. The incidence of respiratory events, such as bradypnea, desaturation < 90%, and apnea requiring rescue maneuvers, were significantly higher with propofol. *Conclusions*: Adding propofol boluses to a remifentanil TCI for GI endoscopy ensures light sedation that may be necessary for anxiolysis but increases respiratory events, even after administration of small-dose boluses. Its safety is acceptable if the procedure is performed in an equipped environment with sedation providers trained to manage respiratory events and drugs titrated to minimal doses.

## 1. Introduction

Procedural or conscious sedation and analgesia in gastrointestinal (GI) endoscopy is widely used to facilitate diagnostic or therapeutic procedures that are painful, anxiety-inducing, or unpleasant, while avoiding general anesthesia and its possible adverse effects [1]. Many anesthesia protocols using midazolam, propofol, or dexmedetomidine for sedation, and opioids for analgesia have been proposed. Since these procedures are mainly performed in an ambulatory setting, drugs with both fast onset for titration and fast offset for early discharge, such as propofol or remifentanil, are especially suitable; we focus on the combination of these two drugs in the present paper.

Combining drugs may reduce the doses of each drug, but the benefits in terms of preventing adverse effects are unclear [2]. Since a synergistic interaction has been demonstrated for both respiratory depression [3] and cardiovascular response [4], such prevention may depend on the dose equilibrium. During propofol-based sedation given at doses high enough to ensure unconsciousness, remifentanil was found to worsen both respiratory depression and hypotension [5,6] but improve patients’ tolerance of the procedure [7]. Conversely, remifentanil-based sedation, combined with propofol boluses targeting bispectral index (BIS) values around 60, can offer adequate analgesia with good tolerance and rapid recovery but no or minimal sedation during the procedure [8]. All these studies suggest that tolerance of the procedure may be a suitable clinical end point, even without loss of consciousness during GI endoscopy, and they support the rationale for remifentanil-based sedation. Since remifentanil has poor anxiolytic and sedative effects, combination with propofol may be necessary; however, the safety of combining remifentanil and propofol at infra-anesthetic concentrations should be examined.

In a previously published bicenter observational study, we investigated the efficacy and safety of remifentanil target-controlled infusion (TCI) given for sedation in various types of procedures [9]. Small boluses of propofol were allowed before starting infusion for anxiolysis, as previously described [10], and during the procedure to suppress response to stimulation. Since the study was observational and followed the practices of each center, the results analysis showed that propofol boluses were given mainly in one center. The propofol doses given were much lower compared with those in all previously published studies (5–50 mg total dose). Therefore, we could identify two groups of patients who had received remifentanil-based sedation, with standardized data recording for all patients.

The purpose of this study is to examine the influence of small boluses of propofol combined with remifentanil TCI, titrated to the tolerance of GI endoscopy, on respiratory and cardiovascular effects, patient satisfaction, and postoperative nausea and vomiting (PONV). By examining these points, we can determine whether to recommend this combination, describe its risks, and define the safe conditions of use.

## 2. Methods

This study was part of a larger bicenter study on the safety, efficacy, and adverse events of remifentanil target-controlled sedation [9]. It was approved by our institutional review board and ethical committee (CSET 2011-1801-0, CPP Kremlin Bicêtre, SC11-020). Here, we focus specifically on the elective GI endoscopic procedures included in the main study. Written informed consent was waived because the study did not affect routine clinical practice; however, oral information was given to the patients, and oral consent was obtained from them. Exclusion criteria were age < 18 years, preoperative bradycardia < 45 bpm, impaired cardiac conduction, predicted difficult ventilation, full stomach, agitation, cognitive dysfunction, refusal to participate, or known intolerance or allergy to remifentanil and propofol.

Upon arrival in the operating room and after insertion of a peripheral intravenous (IV) line, electrocardiogram (EKG), non-invasive blood pressure (NIBP), and pulse oximetry (SpO_2_) monitoring were initiated. The respiratory rate was monitored visually, by thoracic impedance through EKG, or by mask capnography when available. An oxygen supply of 2–3 L/min was delivered to all patients. Certified anesthesiologists were in charge of all patients in both centers.

In center 1, no IV premedication was given, whereas in center 2, 10 mg of IV propofol was given to all patients. Then, in both centers, a remifentanil TCI was started using Minto’s [11] pharmacokinetic model in a Base Primea infusion system, with an initial effect-site target of 2 ng/mL, adapted during the procedure by stepwise increases or decreases of 0.5 ng/mL to reach adequate analgesia.

If agitation or discomfort remained despite several target adjustments, or if the endoscopist’s comfort was impaired, propofol boluses of 5–20 mg were allowed. If inadequate sedation remained, a switch to general anesthesia could be adopted, but procedural sedation was considered a failure.

If bradypnea < 8 breaths per minute or a SpO_2_ drop below 95% occurred, anesthesia providers decreased the target remifentanil concentration and verbally stimulated the patients. If adequate spontaneous ventilation was not restored, jaw thrust and/or face mask ventilation was started until adequate ventilation was reached.

In the post-anesthetic care unit (PACU), in addition to standard monitoring (NIBP, SpO_2_, heart rate, respiratory rate) and nausea and vomiting assessment, the patients were asked about their satisfaction (expressed as a binary variable: satisfied vs. not satisfied), happiness (expressed as happy or very happy vs. moderately unhappy or unhappy), and, finally, whether they would agree with the same method of sedation in case of a future planned procedure.

The parameters recorded for analysis were as follows:Demographic data;Remifentanil and propofol dosing data;Number of patients who completed the procedure without rescue medications or switching to general anesthesia, patient satisfaction, and willingness to have the same technique for a future procedure;Depth of sedation, expressed by the minimal value of Modified Observer’s Assessment of Alertness/Sedation Scale [12]. with grading from 0 = no response after painful trapezius squeeze to 5 = responds readily to name spoken in normal tone;Maximal value on the Pain Numeric Scale (from 0 to 10) during the procedure;Respiratory depression assessed by the minimal value of respiratory rate and SpO_2_ and the number of respiratory depression episodes responding to target adjustment or requiring jaw thrust or ventilation;Hemodynamic stability through minimum and maximum values of mean blood pressure (MBP) and heart rate (HR), as well as doses of rescue medications (atropine, ephedrine) given;Satisfaction, happiness, and readiness for the same procedure with the same analgesic protocol;Incidence of PONV upon arrival in the PACU;

Results are expressed as mean ± standard deviation or number of patients and percentage of patients experiencing an event in the group from the same center. Groups were compared by Student’s t-test for quantitative variables and chi-squared for number of events.

## 3. Results

A total of 89 patients were enrolled in this study, including 29 in center 1 and 60 in center 2. No significant difference was found between centers in the demographic data or type and duration of the procedure Table 1.

The incidence rates of success without rescue treatment were similarly high between the two centers (Table 2). No conversion to general anesthesia was necessary in either center. Anesthesia appeared more unstable in center 2 (where all patients received at least an initial bolus of propofol): The maximal remifentanil target was significantly higher, as was the maximum pain score, but the total and average remifentanil doses were not, and there was a trend toward a higher number of adjustments. Meanwhile, eight patients in center 2 needed at least one rescue bolus of propofol, as opposed to two patients in center 1 (NS). As expected, patients in this group exhibited greater sedation (57% OAA/S 3 or 4 vs. 17%, *p* = 0.0004), but no patient in any group reached very deep sedation (OAA/S δ 2). In the PACU, neither the PONV incidence (17.2% vs. 11.7%) nor patient satisfaction (76% vs. 82%) differed significantly between centers.

Respiratory depression was significantly more frequent and deeper in center 2 (Table 3), where the following findings were observed:62% of patients had a minimal respiratory rate < 8 breaths/min (vs. 21%);26.7% had minimal SpO_2_ < 90% (vs. 6.9%);13.3% needed jaw thrust (vs. 0%).

In the PACU, the respiratory rate and SpO_2_ were significantly lower in group 1, but no respiratory depression severe enough to be treated was observed (Table 3).

Marked respiratory depression increased with propofol dose, but it could also occur with a dose as small as a single bolus of 10 mg (Figure 1). In center 2, maximal blood pressure during endoscopy was significantly higher, and HR was lower, but no patient had critical values to deserve rescue treatments either during the procedure or in the PACU (Table 4).

## 4. Discussion

This study shows that remifentanil-based sedation, administered in TCI and combined or not combined with small doses of propofol, is suitable for GI endoscopy. In the findings, adding boluses of propofol had no significant influence on the remifentanil average requirements or the success of the procedure. Remifentanil induced moderate sedation, but at the same time, it significantly increased the incidence of respiratory depression and the need for rescue jaw thrust and verbal stimulation. Anxiolysis was not recorded, but patient satisfaction was similar with or without propofol.

The rationale for remifentanil-based sedation is to block the responses to endoscopy stimulation, such as motor responses, coughing, or the gag reflex, for upper endoscopy [13,14] or to prevent agitation during colonoscopy. Giving remifentanil alone, Akcaboy et al. [15] achieved pain relief and comfort and observed less sedation, anxiolysis, and amnesia, but there was more nausea than with propofol. In this study, only one patient (2%) had bradypnea severe enough to require assisted ventilation, which is consistent with our results for center 1. Moerman et al. [16] also observed a high incidence of bradypnea (45%), but only two patients in that study had apnea (10%). Conversely, the rationale for propofol-based sedation is primarily to provide anxiolysis, amnesia, and in most studies, unconsciousness [5,6,17]. Yet, because of its lack of analgesic properties, propofol given alone requires high concentrations to prevent a response to endoscopy, with possible central or obstructive respiratory depression [16].

When both drugs are combined, propofol is still often titrated to achieve unconsciousness. Giving propofol up to a predicted concentration between 3 and 5 µg·mL^−1^, combined with remifentanil at concentrations of around 3–4 ng·mL^−1^, Moerman et al. [5] found 30% bradypnea as compared to 0% with propofol alone, with a similar incidence of obstructive apneas (~18%). Five years later, the same group examined a propofol–remifentanil combination with lower remifentanil dosing (Ce 1 ng·mL^−1^ instead of 3–4 ng·mL^−1^), but, again, propofol TCI was titrated until loss of consciousness, with concentrations of around 4–5 µg·mL^−1^. The quality of sedation was lower with propofol alone, with more movements, coughing, and hiccupping. When remifentanil was added, bradypnea, as well as obstructive apnea, was observed. The incidence of respiratory depression was lower when remifentanil was administered by TCI [7].

All the previous studies suggest that the tolerance of the procedure is ensured by remifentanil and that the combination with propofol, up to loss of consciousness, increases the risk of respiratory depression compared with either drug alone. The next issue is to decide whether loss of consciousness is necessary during GI endoscopy or whether lighter sedation may be equally suitable and safer. Both Akcaboy et al. [15] and Moerman et al. [16] showed that patients were similarly satisfied after receiving remifentanil, when they remained conscious, compared to receiving propofol, when they lost consciousness. Deep sedation offers no advantage compared with light sedation in terms of the ease and success of colonoscopy (17). During deep sedation, patients lose both verbal contact and the ability to appropriately respond to incentives to breathe, which may be a safety issue [18]. In another study, LaPierre et al. [13] performed sophisticated pharmacodynamic modeling to determine the couples of concentrations allowing tolerance to gastroscopy without loss of consciousness or respiratory depression. They concluded that this equilibrium may be achieved either with propofol at concentrations of 1.5–2.7 µg·mL^−1^ and remifentanil at up to 0.8 ng·mL^−1^ or with remifentanil at 3–4 ng·mL^−1^ and propofol at up to 0.6 µg·mL^−1^. The study was accompanied by an editorial concluding that the propofol–remifentanil combination is so difficult to manage in spontaneous ventilation that single drug sedation may be preferred [18].

In our study, doses of propofol (boluses of 5–10 mg, total dose < 50 mg) were much smaller compared with the doses in all studies targeting deep sedation. When simulating the corresponding predicted concentrations, they were always less than 1 µg/mL, as exemplified in Figure 2. This resulted in light sedation (OAA/S 3 or 4), consistent with LaPierre et al.’s [13] recommendations. However, our results show that limiting propofol boluses to light sedation does not prevent respiratory depression when given with an opioid. This result could be expected from the interactions in pharmacodynamic modeling performed by Nieuwenhuijs et al. [3], who observed a collapsed response to hypercapnia when giving propofol with remifentanil for concentrations as low as 1 µg/mL and 1 ng/mL. However, the main advantage of light sedation is to make respiratory depression rapidly reversible once detected; this is due to the lack of sedation and fast offset of remifentanil. Bradypnea was reversed in most situations by verbal stimulation and remifentanil target decrease. Therefore, the early detection of bradypnea is crucial for the safety of sedation. This should be based on respiratory rate monitoring, since bradypnea is much more frequent and occurs earlier compared with desaturation [19]. Capnography and thoracic impedance are both suitable.

Among the various procedures performed under sedation with spontaneous ventilation, GI endoscopies showed the largest variability between patients [6,9]. This variability may be due to the large number of procedures defined as “digestive endoscopy”, corresponding to different levels of stimulation—namely, upper versus lower endoscopy, simple colonoscopy versus deep mucosectomy, and so on. Therefore, it is not surprising that titrating sedation for these different procedures requires different concentrations.

The stimulation may vary within the time course of an endoscopy. As an example, Borrat et al. [14] studied upper endoscopy and stated that the most stimulating time was the introduction of the tube from the mouth to the esophagus. TCI is especially suitable for titrating drug delivery to these variable needs, while maintaining spontaneous ventilation [7,20,21].

Despite its advantages, our study has several limitations that should be mentioned. First, it was an observational study, and the criteria to titrate remifentanil or propofol on tolerance or response were imprecise. Second, it compared two sedation protocols (with or without propofol bolus), with each arm recorded in two different centers, where practices may differ in accordance with the anesthesiologists, procedures, or operators. This potential recruitment bias may explain why the maximal remifentanil target was higher in center 2 (with propofol) than in center 1 (without). Third, risk factors for obstructive apnea were not systematically recorded for such short procedures. However, we can at least observe that the number of patients with a body mass index (BMI) of more than 35 kg.m^−2^ or an age over 50 years was similar between centers. Fourth, because of the study design, as well as the small number of patients and events, the study did not allow a precise estimation of apnea risk; rather, the study only showed readers that adverse respiratory effects may occur if propofol is added to the opioid, even with small bolus doses inducing light sedation. Finally, this paper focused on remifentanil and propofol and did not compare them with the gold standards of sedation using benzodiazepines or other opioids. We decided to exclude this comparison, even in the discussion section, because we had no personal data to discuss.

## 5. Conclusions

Low-dose propofol (10 mg) combined with remifentanil TCI in digestive endoscopy yielded a similar incidence of success and level of satisfaction in comparison to remifentanil alone. However, this combination showed a significant increase in respiratory events, although most of these were reversible via verbal stimulation and drug titration. Given the possibility of adverse events, this technique should only be used in a safe anesthetic environment, as recommended in the European Society of Anaesthesiology and Intensive Care guidelines [22], pending propofol being titrated to light sedation and remifentanil to controlled and limited responses during the successive steps of each procedure.

## Figures and Tables

**Figure 1 medicina-58-01285-f001:**
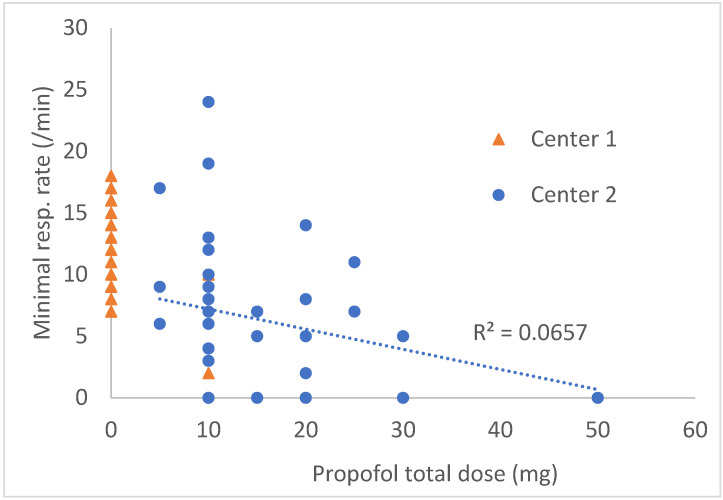
Minimal respiratory rate as a function of total dose of propofol received (1 point/patient). Higher propofol doses were significantly correlated with lower minimal respiratory rates (r = −0.25).

**Figure 2 medicina-58-01285-f002:**
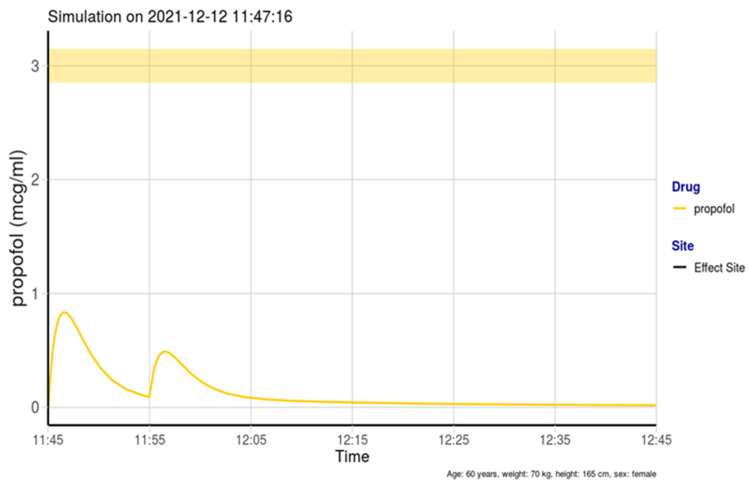
Example of predicted propofol concentration after an initial bolus of 20 mg, followed by a second bolus of 10 mg. Simulation performed with the Stanpump R shareware program (https://stanpumpr.io/ accessed on 12 December 2021).

**Table 1 medicina-58-01285-t001:** Demographics. Mean ± standard deviation or number of patients. NS = not significant.

	Center 1 (*n* = 29) Remi	Center 2 (*n* = 60) Propofol/Remi	*p*
Age (y)	61 ± 12	59 ± 15	NS
Patients > 50 y (%)	82.8%	75%	NS
Gender (male/female)	12/17	37/23	NS
ASA status I/II/III/IV	3/17/7/2	12/32/15/1	NS
BMI kg.m^−2^	27 ± 6	26 ± 5	NS
Patients > 35 kg.m^−2^ (%)	10.3%	6.8%	NS
Coloscopy/upper endoscopy/both	16/2/11	36/4/20	NS
Anesthesia duration (min)	23 ± 21	21 ± 12	NS

ASA: American Society of Anesthesiologists, BMI: body mass index.

**Table 2 medicina-58-01285-t002:** Sedation efficacy and drug delivery data. Mean ± standard deviation or number of patients (%).

	Center 1 (*n* = 29) Remi	Center 2 (*n* = 60) Propofol/Remi	*p*
Number of successes without rescue	27 (93%)	52 (87%)	NS
At least 1 agitation episode	8 (28%)	16 (27%)	NS
Happy or very happy/moderately unhappy or unhappy	22/3	49/11	NS
Ready for a new procedure (Y/N)	24/1	48/9	NS
PONV	5 (17.2%)	7 (11.7%)	NS
Total dose of remifentanil (µg)	208 ± 103	248 ± 132	NS
Average remifentanil rate (µg·kg^−1^·min^−1^)	0.19 ± 0.17	0.18 ± 0.07	NS
Maximum remifentanil target (ng·mL^−1^)	3.6 ± 1.37	5.1 ± 1.5	<0.001
Number remifentanil target adjustments	2 ± 1.7	2.8 ± 2	0.056
Propofol boluses	2 patients had a 10 mg bolus; others had no propofol	8 patients received more than 1 bolus; all others received 1 initial bolus	
Total propofol dose (mg)	0.7 ± 3 [0, 10]	13 ± 8 [5, 50]	<0.001
Maximum pain scale	2.6 ± 2.3	4.3 ± 2.8	0.004
OAA/S 3.4	5 (17%)	34 (57%)	0.0004

PONV = postoperative nausea and vomiting. OAA/S: Observer’s Assessment of Alertness/Sedation scale.

**Table 3 medicina-58-01285-t003:** Respiratory events during the procedure and PACU.

	Center 1 (*n* = 29) Remi	Center 2 (*n* = 60) Propofol/Remi	*p*
Min. SpO_2_ (%)	96 ± 3	93.2 ± 9.5	0.046
Min. SpO_2_ < 90%	2 (6.9%)	16 (26.7%)	0.0295
Min. resp. rate (min^−1^)	11 ± 4	7 ± 5	<0.0001
Min. resp. rate < 8 min^−1^	6 (21%)	37 (62%)	0.0003
Patients requiring jaw thrust	0	8 (13%)	0.039
Mask ventilation	0	5 (8%)	NS
Min. resp. rate in PACU	9 ± 3	13 ± 2	0.0004
Min SpO_2_ in PACU	95.3 ± 1.8	97.3 ± 2.2	<0.0001

Min. = minimum mean ± standard deviation or number of patients (%).

**Table 4 medicina-58-01285-t004:** Cardiovascular monitoring. Mean ± standard deviation or number of patients.

	Center 1 Remi	Center 2 Propofol/Remi	*p*
Min. MBP (mmHg)	90 ± 14	92 ± 17	NS
Max. MBP (mmHg)	108 ± 18	118 ± 24	0.029
Min. HR (bpm)	72 ± 12	64 ± 13	0.014
Max. HR (bpm)	96 ± 20	86 ± 19	0.027
Atropine/ephedrine	0/0	0/0	-
Min. MBP in PACU (mmHg)	87 ± 9	100 ± 19	0.056
Min HR in PACU (bpm)	71 ± 14	65 ± 11	0.002

Min. = minimum, max. = maximum, PACU = post-anesthetic care unit, MBP = mean blood pressure.

## Data Availability

Data available on demand.

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
