# Peer review of "Adding Low-Dose Propofol to Limit Anxiety during Target-Controlled Infusion of Remifentanil for Gastrointestinal Endoscopy: Respiratory Issues and Safety Recommendations"

_medicina, 2022, doi:10.3390/medicina58091285_

Round 1

Reviewer 1 Report

This study evaluated to assess the adverse respiratory and cardiovascular effects of combining propofol small boluses with remifentanil, compared with remifentanil alone and balanced with the quality of sedation and recovery.

Major revision

1.     There was a trend toward more additional doses of propofol in Center 2. Why was this trend observed? Could the additional propofol have influenced the cause of respiratory depression?

2.     Please review the number of patients who required oxygen administration in each center.

Minor revision

1. Was anesthesia management performed by a professional anesthesiologist?

Author Response

We thank the reviewer for the constructive suggestions.

  1. The reviewer is right. Propofol was given more often in center 2 because it was an usual practice in this center to treat anxiety with propofol and this choice has been left open in the protocol. Otherwise, anesthesia protocol and monitoring was similar, and this different use of propofol in both centers suggested to us the present study. Data analysis supported the influence of propofol on respiratory depression. 

2. Oxygen supply information has been added to the revised text.

3. English language has been revised.

4. Of course, in both centers, certified anesthesiologists were in charge for all patients.  

5. The manuscript is now professionaly edited in academic english  in  scribendi editing platform based in ontario canada 

Reviewer 2 Report

Authors provided a well written and scientifically sound manuscript. But I request authors to rearrange the paragraphs in Discussion to reduce the number for better understanding and presentation. 

For example: Line 243 to 256, please make sure appropriate paragraph to consolidate. 

Key strengths of the manuscript are listed below. 

Introduction:

·       It is well written to cover various anesthesia protocols as background. 

·       Problem statement and relevant studies are included.  

Methodology: 

·       Selection and grouping of patients are suitable for the studies to prove the hypothesis. 

·       All appropriate measure were taken to prove the clinical study design to be more effective. 

·       Demographics of patient profile seems to be sufficient

·       Sedation efficacy and administration of drug are recorded carefully in both center 1 and center with minimum number of patients required for statistical evaluation. 

Results & Discussion:

·       All the results are appropriately addressed with appropriate scientific discussion. 

·       Statistical evaluation is applied to get appropriate conclusion about the studies. 

·       Main objectives of the manuscript is to examine the influence of propofol small boluses combined with remifentanil and conclusion about its recommendation is appropriately addressed.  

Author Response

Thanks to the reviewer for listing the strenghts of the study. 

The introduction and discussion sections have been reviewed as required.  

and finally the manuscript has been extensively   edited  in academic english by Scribendi editing service platform based in ONtario Canada 

Round 2

Reviewer 1 Report

I have no additional comments.